# Supervised Machine Learning Enables Geospatial Microbial Provenance

**DOI:** 10.3390/genes13101914

**Published:** 2022-10-21

**Authors:** Chandrima Bhattacharya, Braden T. Tierney, Krista A. Ryon, Malay Bhattacharyya, Jaden J. A. Hastings, Srijani Basu, Bodhisatwa Bhattacharya, Debneel Bagchi, Somsubhro Mukherjee, Lu Wang, Elizabeth M. Henaff, Christopher E. Mason

**Affiliations:** 1Tri-Institutional Computational Biology & Medicine Program, Weill Cornell Medicine, New York, NY 10065, USA; 2The HRH Prince Alwaleed Bin Talal Bin Abdulaziz Alsaud Institute for Computational Biomedicine, Weill Cornell Medicine, New York, NY 10065, USA; 3Integrated Design and Media, Center for Urban Science and Progress, NYU Tandon School of Engineering, Brooklyn, New York, NY 11201, USA; 4Department of Physiology and Biophysics, Weill Cornell Medicine, New York, NY 10065, USA; 5Center for Artificial Intelligence and Machine Learning, Indian Statistical Institute, Kolkata 700108, India; 6Machine Intelligence Unit, Indian Statistical Institute, Kolkata 700108, India; 7Department of Medicine, Weill Cornell Medicine, New York, NY 10065, USA; 8Department of Electrical and Electronics Engineering, Birla Institute of Technology, Mesra, Ranchi 835215, India; 9Department of Metallurgy & Materials Engineering, Indian Institute of Engineering Science & Technology, Shibpur, Howrah 711103, India; 10Department of Biological Sciences, National University of Singapore, Singapore 117558, Singapore; 11WorldQuant Initiative for Quantitative Prediction, Weill Cornell Medicine, New York, NY 10065, USA

**Keywords:** microbial fingerprint, microbial forensics, supervised machine learning, metagenomics, bioindicator species

## Abstract

The recent increase in publicly available metagenomic datasets with geospatial metadata has made it possible to determine location-specific, microbial fingerprints from around the world. Such fingerprints can be useful for comparing microbial niches for environmental research, as well as for applications within forensic science and public health. To determine the regional specificity for environmental metagenomes, we examined 4305 shotgun-sequenced samples from the MetaSUB Consortium dataset—the most extensive public collection of urban microbiomes, spanning 60 different cities, 30 countries, and 6 continents. We were able to identify city-specific microbial fingerprints using supervised machine learning (SML) on the taxonomic classifications, and we also compared the performance of ten SML classifiers. We then further evaluated the five algorithms with the highest accuracy, with the city and continental accuracy ranging from 85–89% to 90–94%, respectively. Thereafter, we used these results to develop Cassandra, a random-forest-based classifier that identifies bioindicator species to aid in fingerprinting and can infer higher-order microbial interactions at each site. We further tested the Cassandra algorithm on the Tara Oceans dataset, the largest collection of marine-based microbial genomes, where it classified the oceanic sample locations with 83% accuracy. These results and code show the utility of SML methods and Cassandra to identify bioindicator species across both oceanic and urban environments, which can help guide ongoing efforts in biotracing, environmental monitoring, and microbial forensics (MF).

## 1. Introduction

The decreasing cost of DNA sequencing has made the study of microbial and other species increasingly possible, ranging from the human-built environments of cities to isolated and extreme natural environments. Large-scale genomics projects such as the Earth Microbiome Project (EMP) ([1]) and the Human Microbiome Project (HMP) [2] have pioneered the comparative study of microbial species across targeted ecosystems. In addition, the Tara Oceans Study [3], the Metagenomic and Metadesign of Subway and Urban Biomes (MetaSUB) Consortium [4,5], and the Extreme Microbiome Project (XMP) [6,7,8] have all broadly expanded the collection of diverse environmental microbial samples, revealing biodiversity and genetic differences between environments at a global scale.

The advancement of DNA sequencing technologies, along with global efforts generating large-scale molecular and annotated data sets, played an integral role in expanding traditional forensic methods to those that can utilize DNA-based material evidence in criminal investigations [9,10]. Historically, there is evidence of microbes providing ancillary evidence in criminal cases, including bioterrorism, pathogenic outbreak, and transmission, post-mortem analysis, microbial tracing as well as identification of humans through microbial “fingerprints”, which are signatures characteristic of a particular environmental location or niche [11,12]. Previous work has also revealed that using large-scale genomics datasets can provide the statistical confidence needed for forensics [13]. Outside of human-associated ecosystems, microbial forensics (MF) could feasibly be used—in the oceans, for example—for tracking deleterious algal blooms or organisms with unknown migration patterns [14]. This would be complementary to existing approaches in which oceanographers use discarded host DNA to ascertain the presence of a given species [15].

The goal of microbial forensics is to be able to identify the point of origin, provenance, or likely history of any given sample using microbial fingerprints: distinct microbiomes that can identify their place of origin [16]. Prior work in MF has revealed that there is a distance decay relationship between soil microbiome and geolocation, which could be exploited for biotracing [17]. Work done by Jesmok et al., 2016 and Chase et al., 2016 demonstrated the successful assignment of places of origin using microbial profiles with the help of supervised classification methods, such as k-nearest neighbor (k-NN) and support vector machines (SVM) [18,19]. Microbial fingerprinting methods have also been used to implement microbial surveillance, as demonstrated by Sanachai et al. (2016), who identified the site of origin from soil microbes in the sole of the shoe using 16S recombinant DNA (rDNA) profiles [20]. Additionally, Segata et al., 2011, developed Linear Discriminant Analysis (LDA) effect size (LEfSe) method and used LDA-based methods for finding biomarkers, including phenotypic indicators from 16S samples [21]. Using ensemble machine learning approaches in application to forensics, Kim et al., 2015 exhibited that gene expression profiles from the bacterium *Escherichia coli* can be used for determining environmental conditions, including abiotic and biotic components, and when additional genome-scale transcriptional information was provided, the classifier outperformed the previous results [22]. Ideally, metagenomic datasets could be used to develop discriminating microbial signatures or fingerprints, but there is a lack of universally accepted tools for identifying bioindicator species from metagenomics data [10,23].

Though it is clear that microbial interactions play a significant role in many environments, methods associated with applications within microbial forensics (MF) remain limited as the field is still in a nascent stage [11,24]. While promising, microbial forensics has historically used reductive or targeted methods for data processing, including Operational Taxonomic Units (OTUs), which focus solely on sequence variation in targeted regions such as 16S and 18S ribosomal RNA (rRNA) or the internal transcribed spacer (ITS). By leveraging whole-genome sequencing data generated by short-read, long-read, and linked-read methods, a more comprehensive view of samples can be leveraged for biotracing purposes. Even though we can generalize all microbial sequence (16S, long-reads, WGS) data to be a sparse vector quantifying frequencies (relative/absolute) within Euclidean space [25] or large kmers [4], the variation of sequencing technologies makes it computationally challenging to quantify and interpret cross-platform and cross-study data [26]. Moreover, current forensics tools do not account for differences due to variations in the sampling method [10], extraction methods [27], and possible noise due to amplification, sequencing, or contamination. Furthermore, most microbial forensics studies focus on large-scale environmental profiling of natural sources and only recently have data been related that focuses on the urban microbiome [4,28].

Here, we present *Cassandra*, an SML tool to address these unmet needs of metagenomics-based microbial forensics, which can predict bioindicator species from shotgun sequence datasets. Microbial species selected by Cassandra over multiple iterations were considered “bioindicator microbial species” for microbial prioritization, meaning that they were the top taxa unique or enriched across locations. To build this tool we first integrated six pre-processing methods to address the common challenges of sparsity and composition affecting data representation of sequence data processed by commonly used metagenomics toolkits in the pipeline. Our open-source tool, *Cassandra*, is a random-forest-based algorithm that deploys machine learning to identify bioindicator species using quantitative microbiome profiles as features. Using the data available through the MetaSUB Consortium—which includes extensive metadata including geolocation, the surface of origin, and temperature—we first explored the presence of microbial fingerprints by testing multiple approaches, including SML methods, traditional clustering, and statistical approaches, using a protocol developed for MetaSUB [4,29]. We were further able to identify multiple species that could be used as bioindicator species with high confidence using Cassandra to further confirm the tool’s utilization for predicting bioindicator species, we tested on the orthogonal dataset (Tara Oceans using OTU-based microbial profiles) (https://ocean-microbiome.org/).

## 2. Methods

### 2.1. Dataset Details

The MetaSUB Consortium aims “to create geospatial metagenomic and forensic genetic maps” (MetaSUB, 2016). The organization has been collecting annual metagenomic data since 2015, providing the largest to-date, built-environment microbial study. The dataset provides a rich resource of urban microbiomes covering more than 60 cities located across 6 continents, and it also provides longitudinal metagenomics data for 16 cities. Prior MetaSUB work has explored the diversity of microorganisms in a built environment consisting of mass transit systems, including subways, buses, and waterways, across various geographical locations [4]. The dataset used for this analysis contains species-level resolution from 4305 samples processed by KrakenUniq from 60 unique cities (including negative controls and positive controls), 9 continental divisions, and surface materials (including some controls).

The Tara Oceans Foundation has been exploring the water for scientific knowledge since 2003. We validated our tool with the publicly available Tara Oceans dataset consisting of microbes processed using the 16S framework [3], with the supplementary dataset was downloaded from the publication. We used the metagenomic data [referred to as MetaG (https://ocean-microbiome.org/)], consisting of taxon classified from gene-coding sequences isolated. We trained our model separately on Phylum, Class, Order, Family, Genus, and OTU from the reported taxa classification. We trained on the feature “OS_region,” which divided the dataset based on locations, including, the Red Sea, Mediterranean Sea, Indian Ocean, South and North Atlantic Ocean, North and South Pacific Ocean, Arctic Ocean, and the Southern Ocean.

### 2.2. Data Preprocessing and Adopted Classification Methodologies

Since different metagenomic tools including QIIME [30], MetaPhlan [31], Kraken [32], and others report taxonomic assignments to use different methods, the species data produced have different quantifiable attributes with respect to compositionality, sparsity [26,33]. Given that microbiome datasets can be of multiple formats, we tested the 6 methods for preprocessing to reduce the biases and quantitative challenges.

*binary*: Binarizing to 0,1 based on a threshold value (default = 0.0001).*clr*: Transformation function for compositional data based on Aitchison geometry to the real space.*multiplicative replacement*: Transformation function for compositional data uses the multiplicative replacement strategy for replacing zeros such that compositions still add up to 1.*raw*: No preprocessing.*standard scalar*: Standardizing the data by removing the mean and scaling to unit variance.*total-sum*: Converts all the samples to relative abundance.

We selected 10 models from 7 different families based on their successful deployment in earlier metagenomic studies [34,35] and further developed a Voting Classifier using 3 of the top performing models. Specifically, this included: (1)Logistic Regression, which classifies estimating the probability of an event occurring based on a priori knowledge.(2)Linear Discriminant Analysis (LDA), a generalization of Fisher’s linear discriminant for separating multiple classes based on linear combination of features.(3)Ensemble Methods, which are an amalgamation of multiple models to produce an optimal classification (Random Forest Classifier, Extra Tree, AdaBoost).(4)Tree-based Classifiers including Decision Tree that use branching methods to evaluate the outcomes for classification.(5)K-nearest neighbor (kNN), a Supervised Learning method which classifies it by a vote of K nearest training objects as determined by some distance metric.(6)Bayesian Classifiers, including Gaussian NBC which update their posterior probability after ingesting new kinds of data.(7)Support Vector Machines (SVM), which separates classes by using a maximal margin hyperplane based on nonlinear decision boundary (Support Vector Classifier and Linear Support Vector Classifier (lSCV)).

The Voting Classifier, another ensemble classifier was created using multiple models and predicted on the basis of aggregating the findings of each base estimator, which was created using RF, lSVC, and Logistic Regression. Table 1 shows the parameters tuned for machine learning prediction; the user-defined parameters can be set manually. If it is blank, it refers to the default setting for the ML model.

A common problem in ML predictions is overfitting the test data. To eliminate overfitting, we added random Gaussian noise with a mean 0 and standard deviation ranging between 10E−10 and 10E+3 which was done using the “--noise” value set to be TRUE. The result includes precision, recall value, and top 1, 2, 3, 5, and 10 accuracies (the number of times the correct label is among the top n labels predicted). The accuracy is the accuracy for a subset computer for multilabel classification based on the Python3 SciPy package. The tools also report the confusion matrix for multiclass multilabel classification.

### 2.3. Addressing Overfitting by Cross-Validation

Beyond the standard methodologies listed in Table 1, it is also necessary to accommodate and correct small human and machine errors, such as sampling variability, DNA heterogeneity, and missing metadata. To address these issues, we test for overfitting of the data by methods including the addition of Gaussian Noise, cross-validations (k-fold cross-validation), and cross-parameter analysis (leave-one-group-out). The k-fold cross-validation makes the model more generalizable, by estimating the model performance in new sets of data created by different model splits, thus providing a more accurate estimation of accuracy. Leave-one-group-out uses a third-party provided group to create a split, where the group information encodes arbitrary domain-specific stratifications of the samples. Both the cross-validations report the best accuracy, average accuracy scores, and standard deviation in accuracy. Machine learning approaches can be used for spatial extrapolation. In Section 3.3, we first use k-fold cross-validation to classify the accuracy of parameters. We then use leave-one-group-out cross-validation, choosing city and continents as the group, while looking into the prediction accuracy for the parameters. Here, we use the cross-validation methods to validate critical parameters that could be affected by interpolation accuracy.

### 2.4. Cassandra Predicts Bioindicator Species Providing Explainability and Interpretability of Datasets

Cassandra was developed as a factor-analysis method to find the bioindicator microbial species (Figure 1A,B). This can be defined as the species chosen by random forest to differentiate between the various sub-environments [36]. Notably, the top five methods compared here (Table 2) displayed high accuracy and low standard deviation, which matches some prior work. Bioinformatic-based benchmark studies by Couronné et al., 2018 have shown Random Forest classifiers generally perform better than Logistic Regression in 69% of the datasets [37]. Other studies designed for metagenomic datasets by Statnikov et al., 2013 supported RF, SVM, kernel ridge regression, and logistic regression are the best methods for microbiome datasets [38]. We chose to use a Random Forest because it yields output that is simple to interpret, stable against Gaussian noise, does not overfit in unbalanced datasets, and the standard deviation between the various preprocessor methods is extremely low.

The input to Cassandra takes the metadata feature of interest, like geolocation, temperature, and the minimum accuracy for prediction expected by the user for the dataset. Using standard machine learning protocols, Cassandra first does an 80–20 train-test split, creates a new tree based on the training set, and trains the model. It evaluates the accuracy of the test set. If the desired accuracy is achieved, Cassandra selects the run and reports the attributed features (microbes) for the run. The algorithm keeps running till we have 1000 instances to achieve consensus (which can be modified by users) with the desired accuracy. The weight of all the species is reported for each of those instances, which helps us understand higher-level interaction between species and the geolocation. Furthermore, another output file is generated which selects the top n-species of interest (which can be user modifiers) of interest which are bioindicator species.

If the accuracy cutoff provided is unachievable in 15,000 runs, Cassandra archives arbitrary accuracy ranging from 0 to less than the desired value, based on the dataset. Hence, exploratory analysis for classification was run based on Danko et al., 2021 protocol, based on which we selected the accuracy parameter [4]. We used the entire MetaSUB dataset to predict bioindicator species with >80% accuracy for cities and >90% accuracy for continents and >85% accuracy for predicting OTUs for the Tara Oceans dataset.

## 3. Results

### 3.1. Microbial Fingerprints can Be Observed from the MetaSUB Dataset Both at the City and Continent Levels

For evaluating the potential of geographical profiling to identify the city and continent of origin using the microbiome, we compared the performance of 10 models on the MetaSUB dataset (Figure 2). The best classifiers for city prediction were Logistic Regression, Linear Support Vector Classifier (LSVC), Support Vector Machine (SVM), Extra Tree, and Random Forest (RF), with most showing >80% accuracy (Table 2 and Figure 2A). Using the top 3 models, namely LSVC, Logistic Regression, and RF, we developed an ensemble voting classifier, which was able to reach an accuracy of 89.51% and had a standard deviation of 0.0175. Overall, these data showed that microbial fingerprints are a good indicator for determining geolocation. Notably, classifications for continents using shotgun metagenomics data outperformed those for cities with (Figure 2B) Linear Regression, Logistic Regression, and Voting Classifiers, reaching over 90% accuracy.

However, metagenomics data can have technical noise and other biases introduced by experimental protocols, sequencing methods, or data-cleaning approaches, and this can create challenges for classifier algorithms [39]. We simulated noise in the natural environment by adding Gaussian noise to gauge the impact on the models’ performance (Figure 2C). Most models maintained their performance metrics, even with the addition of random Gaussian noise, with a mean change of 0 and standard deviation ranging between 10E−10 to 10E+1), indicating our model does not likely suffer from overfitting, but still demonstrates the importance of preprocessing and model selection (Figure 2D). In general, we observed that all the classifiers performed better than random chance (1/no_of_distinct_classes = 1/60 = 0.0167 or 1.67% accuracy) which was achieved with a training time of fewer than 10 min for each instance (Figure 2C).

### 3.2. Adding Interpretability and Explainability to Machine-Learned Microbial Fingerprints by Characterizing Bioindicator Microbes with Cassandra

Prior work by Danko et al., 2021 showed that RF models can be used for city classification from shotgun sequence data, and as such, we decided to use an RF method to develop Cassandra, a new tool to identify bioindicator species associated with geolocations [4]. Cassandra requires user input of desired accuracy, a parameter that defines the minimum accuracy expected for uniquely identifying cities. As we have observed from previous experiments, RF could achieve an accuracy score of around 80% for classifying cities. Thus, we set Cassandra to achieve 80% accuracy for classifying our desired biomarker species. We also selected *clr* preprocessing for our dataset, as it was most stable against noise, as indicated in Figure 2C. Cassandra took 1108 runs for a feature selection to achieve an average accuracy of 81.3% across 1000 iterations.

We next examined the output of Cassandra after this processing, which details the weight associated with each microbe for classification with minimum accuracy. We found that microbial abundance (the amount of a given microbe) and microbial prevalence (the number of samples where the microbes were found) had a linear relationship (Figure 3A). Furthermore, we found the bioindicator species displayed a strong correlation with microbial abundance and prevalence (Figure 3B,C). Previous studies (Danko et al., 2021) have shown a bimodal, wide distribution of taxa prevalence across the MetaSUB dataset, and, as expected, the abundance of the top 50 bioindicator species showed a wide variance (Figure 3D) as well [4]. Out of the 31 core microbial taxa in the MetaSUB dataset, we observed 4 of them in the top 50 bioindicator species, namely: *Streptococcus mitis, Brevundimonas sp. GW460, Brevundimonas naejangsanensis, Cutibacterium acnes*.

These results further revealed: new insights about bioindicator species for the MetaSUB dataset: the algorithm selected for a combination of abundant microbes (which have a differential presence across cities), unique microbes associated with a given region, as well as a few core microbes with differential abundances across cities. Hence, both the abundance and presence of microbes are utilized by machine learning tools for estimating unique city-based fingerprints. When we compared the top 50 species of microbes selected for predicting continents (average accuracy attained by Cassandra: 90.3%), we found that around half (22/50) of species were shared with the top 50 species selected for predicting cities (Figure 3E), but the rank of the species provided by Cassandra differed at both continent and city levels.

### 3.3. Feature Interpolation for Microbial Forensics can Be Achieved Using Microbial Data

Using the MetaSUB dataset, we next examined the ability to predict other forensic features, including surface material (e.g., metal, glass, plastic), temperature, climate, and other clues of sample provenance. To develop these methods, we trained the MetaSUB species data on eight unique features, including surface type, sampling type, coastal city, climate, and others (Table 3). Each feature was analyzed independently using 3 cross-validation methods (k-fold, leave one group out for the city, and one for the continent), which repeatedly uses part of the samples for learning the model, and the remainder for validating the predictions (Methods 2.3).

Three models were separately trained for each feature and cross-validation methods using the following (classifier + preprocessing) methods: random forest with standard scalar, the voting classifier (mixed model) with clr, and logistic regression with multiplicative replacement. After compiling all outcomes, we observed that surface material always outperformed other metadata types (Table 3). In general, microbial data can also classify other associated features with geolocation, like elevation, coastal association, and climate. To confirm this, we also tested classification with LOGO, where we also saw some features like surface material (coarse) outperforming k-fold validation.

### 3.4. Modeling the Microbial Fingerprint of the Tara Oceans Dataset

We next tested our ability to detect microbial signatures in non-human-associated ecosystems, specifically, the Tara Oceans dataset. We trained separately for all the separate taxon levels to classify the oceanic location from which the sample is collected. In general, we found that we were able to predict all taxon levels, with a precision and recall of greater than 65% (Figure 4), including Operational Taxonomic Units (OTUs). For class, genus, and OTU, the accuracy was greater than 80%. We then used OTUs to train Cassandra, which achieved 86.5% accuracy (the top 15 bioindicators’ OTUs are depicted in Figure 4B).

## 4. Discussion

In this study, we describe the utility of SML tools and further developed Cassandra in identifying and attributing bioindicator species in oceanic and urban environments that can help guide future microbial forensics efforts. This investigation demonstrates that, even from varied mixtures of microbial communities, we can uniquely predict the provenance of a given metagenomic sample by its microbial fingerprint. Deploying the MetaSUB protocol [4], we were able to classify the continent and city of origin from a given WGS metagenomic sample from the global dataset, with 94% and 89% accuracy, respectively, and the sea or ocean of origin with 83% accuracy when using OTUs from the Tara Oceans dataset. Thus, Cassandra shows that these methods can be utilized for metagenomic forensics, for example, in the comparison of samples from unknown and questioned origins.

In this study, we explored whether microbial signatures from known sources and locations can be utilized for predicting the origin of an unknown sample. We were able to accurately extrapolate location-based metadata—such as identifying an urban environment or proximity to the coast—with an accuracy as high as 87.1% (Table 2) from surface ontology. For most cases (with exception of Surface Material and Surface Ontology), extrapolating features at the city level outperformed at the continent level (Table 3). Well-curated metadata for microbial studies at a global scale make such associations possible, and are required to provide ancillary evidence for the purpose of using these methods in microbial forensics, but could also be used to reduce boundary negotiating artifacts amongst different study cohorts [40]. Irrespective of growing evidence suggesting the usage of microbial forensics to complement traditional forensics methods, a more detailed categorization of the environmental association of microbes is required including microbiome-disease association [41,42], along with more publicly available tools for analysis and more reproducible quality of research [43]. 

The use of Cassandra as a tool for the identification of species can be extended to multiple circumstances including pathogenic propagation as a biodetector [44,45], and the spread of antimicrobial resistance [46,47]. Looking ahead, future integration using tools like MetaMeta [48] can homogenize analysis for cross-study data integration. Species classified by Cassandra can be further used to monitor pathogenic outbreaks as well as the spread of AMR using additional resources. Databases like Global Biodiversity Information Facility (GBIF) [49], TerrestrialMetagenomeDB [50], and Microbe Directory (MD2) [51] can be used as ancillary information for defining phenotypes, and other attributes of the bioindicators species [52]. Thus, the use of this tool can help improve source tracing and phylogenetic reconstruction to determine disease transmission in geographical locations in the situations of biocrime and biothreat. 

A key objective of microbial forensics is to utilize microbial analyses and other evidence to resolve a sample’s provenance, and geospatial attribution is often more challenging than simply identification. For example, increasing the resolution of a location estimate gets harder as the desired specificity increases (e.g., city vs. continent), and cities have varying degrees of autochthonous and unique species. Moreover, noise elimination, auto-correlation, and sample mapping are dependent on many factors, and in this paper, we have tested a range of options for data cleaning and their impact on SML tools. Microbial forensics deals with a wide range of microorganisms, including viruses, fungi, parasites, bacteria, and small eukaryotes, and future work can tease out more interactions between species as well.

Using metagenomic samples, this model could provide more power to microbial forensics analyses by helping with the characterization of the origin of a sample based on its microbial signature, such as at the scene of a crime. Although a daunting endeavor, Cassandra could be trained to identify the place of the crime, and the place of the death based on the geographic distribution of the microbial communities. While Cassandra can have myriad applications, proper interpretation necessitates the requirement of proper guidelines, orthogonal validation, and statistical ranking.

## 5. Conclusions

Microbial forensics has a wide range of applications, and work in the field is still in its infancy. Nonetheless, forensics approaches leveraging microbes have been used in tracking crimes [53], the provenance of samples in a city (e.g., MetaSUB), and also for sexually transmitted diseases like HIV and HCV (Network, B). Efforts led by the Human Microbiome Project were integral in understanding the microbial fingerprint across different body sites [2,54], and this work is now being leveraged as well in microbial forensics, such as the human skin signature in the MetaSUB dataset. New studies conducted by Franzosa et al., 2015 explored the use of the human microbiome as a personal identifier, which also can aid microbial forensics [55]. The authors reported that body site-specific microbial profiles of individuals could be used to match an individual, even several months later. Other applications include exploring the thanatomicrobiome, in determining the time elapsed since microbial death [56]. Further applications include tracking the microbiome before and after exposure to diseases like SARS COV2 [57], and determining changes in the microibial composition following dietary alterations [58,59,60].

Microbial forensics may have future uses in other areas, such as the identification of an individual’s past exposures or countries/geographical locations previously visited [61], which can bring a temporal dimension to microbial tracking. This information could be used to highlight whether a subject is a person of interest for law enforcement services, as well as a means by which to eliminate suspects, which would make it a powerful tool that could be used as ancillary evidence in criminal cases. However, such use requires caution, as it could contribute to errors, socioeconomic discrimination, or stigmatization, and the results would need to be verified, as with all other methods in forensics. Nonetheless, the tools, data, and methods are finally emerging to make such methods of tracking and provenance a reality.

## Figures and Tables

**Figure 1 genes-13-01914-f001:**
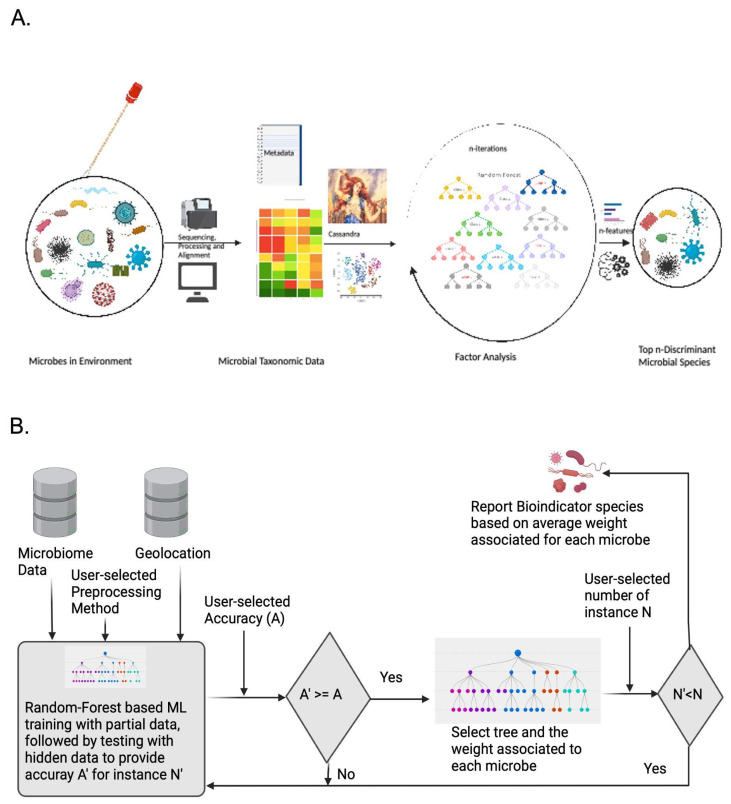
Working principle of Cassandra. The Random Forest-based method is designed to select bioindicator species for applications to microbial forensics. (**A**) Diagramatic schematic showing a conceptual interpretation of how Cassandra selects top bioindicator species for discriminating location from microbial data and geolocation and (**B**) Algorithmic Schema that Cassandra uses for reporting species of interest.

**Figure 2 genes-13-01914-f002:**
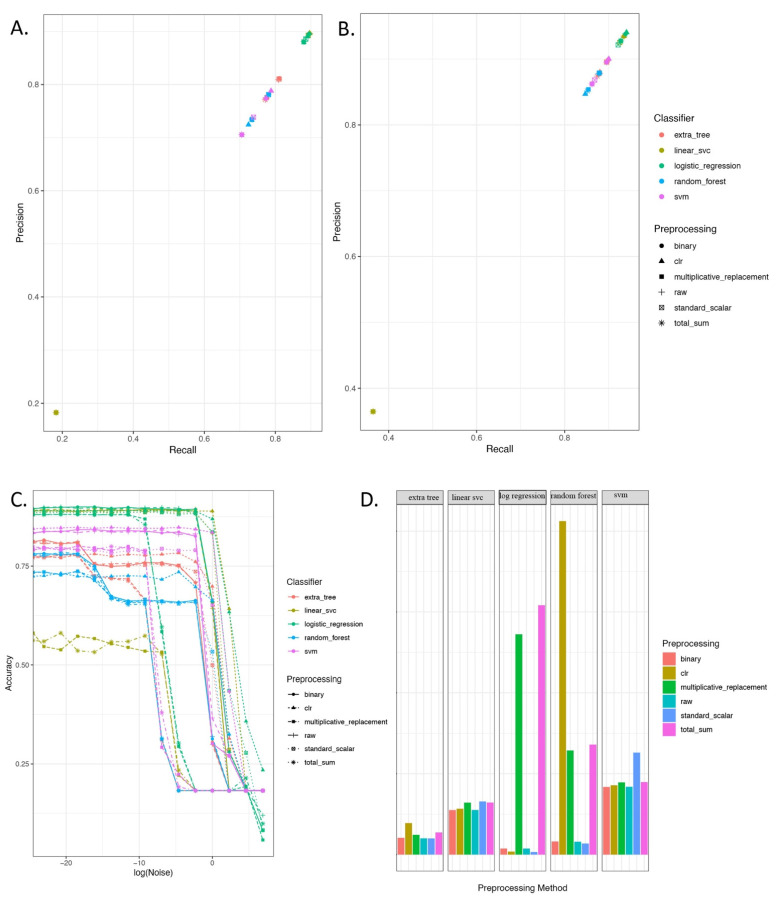
Performance of the ML tools for predicting geolocation from MetaSUB dataset. Top 5 methods to detect microbial fingerprints of cities with high precision and recall for (**A**) city. (**B**) continents. Micro-averaging (used for un-balanced classes in NumPy) has been used for calculating the precision and recall values to account for class imbalances. (**C**) Gaussian noise (used to mimic metagenomic noises) for the best preprocessing method for each model to predict city (**D**) Training time required for city classification.

**Figure 3 genes-13-01914-f003:**
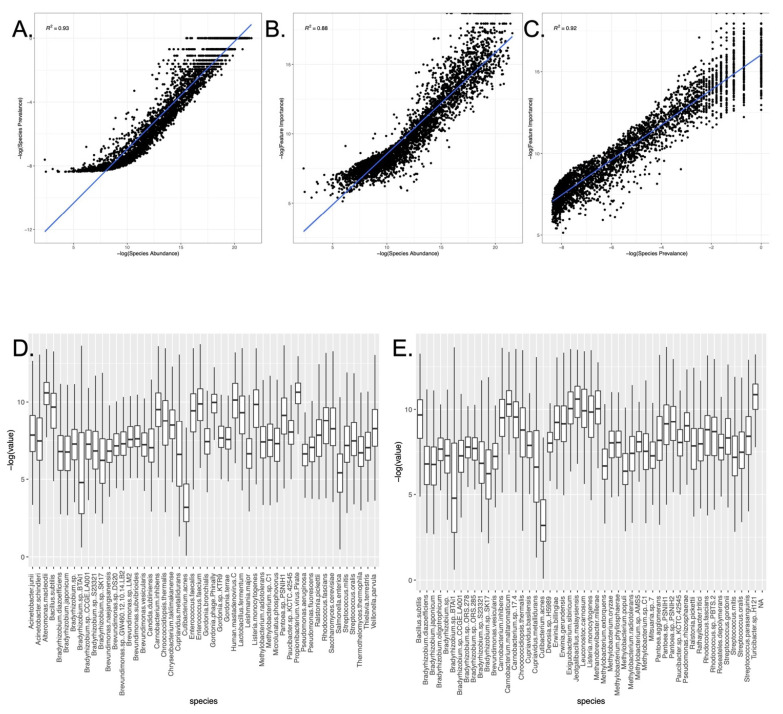
Bioindicator Species and their association with other metrics. (**A**–**C**): The feature importance (weight assigned to bioindicator species) of the microbes as a bioindicator for cities, species prevalence (number of samples the species is present in), and species abundance (relative abundance of species across all samples) shows a linear relationship when plotted against one another (**D**) Boxplot depicting the abundance of the top 50 bioindicator microbial species for cities in the original MetaSUB data (**E**) Boxplot depicting the abundance of the top 50 bioindicator microbial species for the continent in the original MetaSUB data.

**Figure 4 genes-13-01914-f004:**
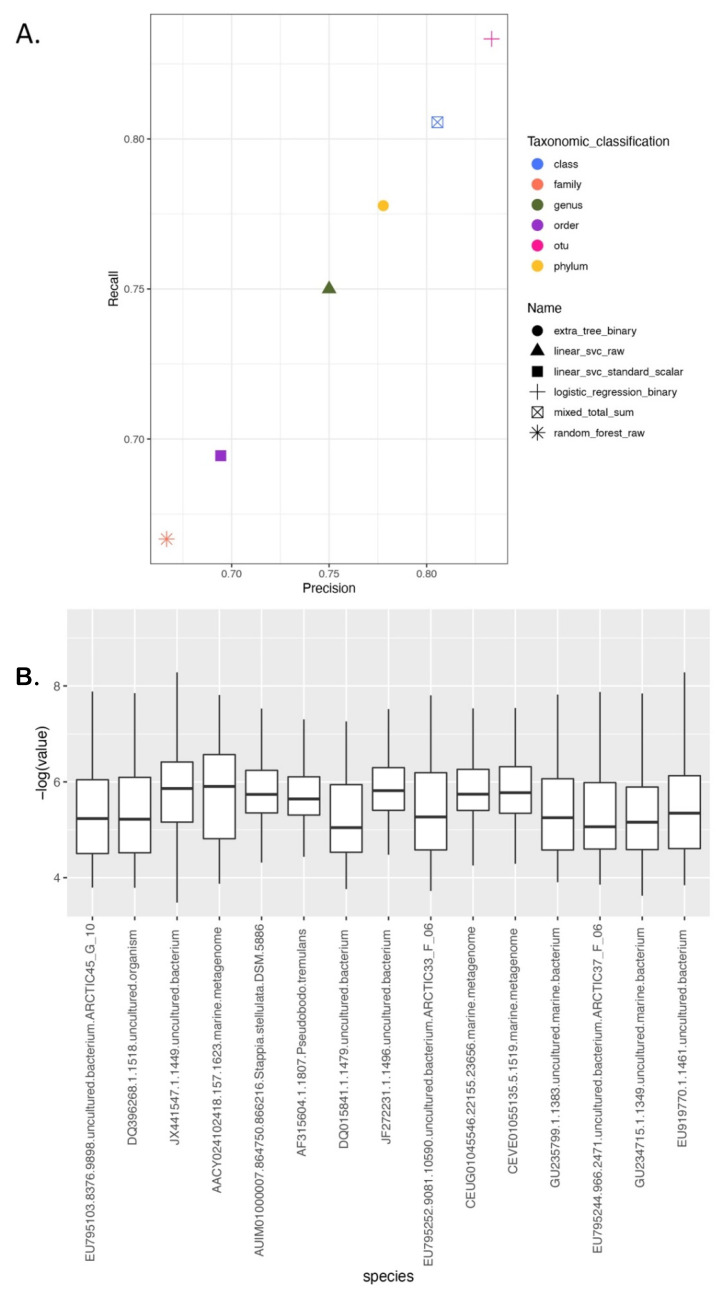
Validation using the Tara Oceans dataset shows the presence of microbial fingerprints in the water sample: (**A**) Precision vs. Recall for the best model for prediction at each category to predict geolocation based on ocean region. We observe that irrespective of being the same dataset when we try classifying based on different domains, the best model/preprocessing differ, even if they are able to achieve similar accuracy. (**B**) Top 15 OTUs selected by Cassandra from Tara datasets along with their weight assigned by Cassandra.

**Table 1 genes-13-01914-t001:** Machine Learning parameters used for training. The method name and the code parameters are listed based on the Python3 SciPy package. If left blank, it means default parameters were used.

Machine Learning Methods	Defined Parameters
AdaBoost	n_estimators = 1000 (User Defined), learning_rate = 4
Decision Tree	-
Extra Tree	n_estimators = 1000 (User Defined), criterion = ‘entropy’
Gaussian Naive Bayes	-
K-nearest neighbors	n_neighbors = 21 (User Defined)
Linear Discriminant Analysis	-
Linear Support Vector Classifier	kernel = ‘linear’, probability = True
Logistic Regression	solver = ‘lbfgs’, C = 1e5, max_iter = 1,000,000
Voting Classifier	voting = ”soft”
Random Forest Classifier	n_estimators = 1000 (User Defined), criterion = “entropy”, bootstrap = True, warm_start = True
Support Vector Machine Classifier	gamma = ‘scale’, decision_function_shape = ‘ovo’, kernel = “rbf”, probability = True

**Table 2 genes-13-01914-t002:** Performance of the top 5 models for city classification: The accuracy and standard deviation between the model accuracy for the 5 five models.

	Preprocessing Method						
Classifier	Binary	CLR	Multiplicative Inverse	Raw	Standard Scalar	Total-Sum	Standard Deviation
**Extra tree**	**0.812**	0.775	0.776	0.809	**0.812**	0.771	0.018328
**Linear SVC**	0.889	**0.893**	0.581	0.89	0.883	0.562	0.1499
**Logistic Regression**	**0.8952**	0.89	0.879	**0.895**	0.8856	0.881	0.00617
**Random Forest**	**0.781**	0.734	0.778	0.778	0.779	0.735	0.02126
**SVM**	0.834	**0.845**	0.79	0.833	0.792	0.798	0.02226

**Table 3 genes-13-01914-t003:** Cross-validation methods for data interpolation: For 8 features, we perform cross-validation based on 10-fold and leave-one-group-out split based on cities/continent as groups. The table explains the features and reports the average accuracy and standard deviation for the different splits for the best method among the above 3 mentioned. In general, we find the accuracy for predicting a feature from an unknown city higher than that from an unknown continent.

Feature	#Number of Variables	Desc. of Feature	10-Fold Av. Accuracy (%)	Leave One Group out (City) Av. Accuracy (%)	Leave One Group Out (Continent) Av. Accuracy (%)	Random Chance (%)
**City climate**	13	Categorized climate Types	93.0 ± 0.0168	34.5 ± 0.3537	12.7 ± 0.1338	7.69
**Coastal**	3	Coastal city and altitude for non-coastal city	90.9 ± 0.0123	59.1 ± 0.3197	40.8 ± 0.1323	33.33
**Coastal City**	2	Binary values	90.9 ± 0.0108	52.4 ± 0.2409	40.5 ± 0.1009	50
**Location Type**	68	Location for collection	85.7 ± 0.0216	44.2 ± 0.3358	40.6 ± 0.3221	1.47
**Surface**	460	Type of surface	47.2 ± 0.0177	8.7 ± 0.1107	3.8 ± 0.0253	0.22
**Surface Material**	115	Standardizeding the “Surface” parameter	54.3 ± 0.0178	24.4 ± 0.1920	32.5 ± 0.2603	0.87
**Surface Ontology (Fine)**	6	Type of surface: stone, biological, etc	66.2 ± 0.0248	50.3 ± 0.2495	61.8 ± 0.1532	16.67
**Surface Ontology (Coarse)**	3	Surface permeability/ Control	85.4 ± 0.0127	87.1 ± 0.1722	83.0 ± 0.0980	33.33

# refers to Number of.

## Data Availability

Code for Cassandra is available here: Cassandra: https://github.com/Chandrima-04/Cassandra. The machine learning methods mentioned in the paper are available as a GitHub package: Meta_Pred: https://github.com/Chandrima-04/meta_pred. The data is based on MetaSUB Consortium dataset and Tara Oceans dataset. Data generated and codes are available in https://github.com/Chandrima-04/Gene_ML_Forensics. All the jobs were run in the Weill Cornell Medicine cluster Curie maintained by the Scientific Computing Unit (https://scu.med.cornell.edu/site). The reported time for running machine learning models is subject to resource availability at a given instance and might require additional time if run locally.

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
