# Peer review of "Supervised Machine Learning Enables Geospatial Microbial Provenance"

_genes, 2022, doi:10.3390/genes13101914_

Round 1
Reviewer 1 Report
General Comments
This manuscript is well written and interesting for the wide applications described. The Authors developed Cassandra, a classifier based on machine learning that identifies “bioindicator” species to aid identifying geo-specific microbial fingerprints. To do so, the Authors integrated previously developed methods in a new algorithm.
Here are some general comments.
The Methods and the Results should be improved in the details and “accuracy”.
The discussion covers the main topics related to the application for microbial forensics using ml approaches. However, it does not go deep in the explanation and argumentation of some of the results. For instance, the Authors found that
“microbial abundance (the amount of a given microbe) and microbial prevalence (the number of samples where the microbes were found) had a linear relationship (Figure 3A). Also, we found the bioindicator species displayed a strong correlation with microbial abundance and prevalence”.
Are these correlations meaningful and reasonable from an ecological point of view?
Similarly, the evidence that about half of the top 50 species of microbes selected for predicting continents are the same for cities.
Specific Comments
L33: higher-order microbial interactions
It is not clear what do the Authors mean
L42 and L54 “The decreasing cost of DNA sequencing” and “The advancement of DNA sequencing technologies”
I would not separate the decreasing costs and the advancements, they go together
L61
I suggest here to chose additional references, beside the one indicated which is a preprint
L71 MF
Make explicit the acronym the first time that microbial forensic is mentioned
L96 Leverage
Please, rephrase to avoid repetitions
L120 “We were further able to identify multiple species that could be used as bioindicator species with high confidence using Cassandra (81.3% for cities, 90.3% accuracy for continents using WGS-based meta- genome profiles).”
Anticipation of the results. Some Authors prefer to anticipate the main findings in the intro section. However, in this case, I would leave the introduction as a true background section.
L123-126
Please, rephrase to make it clear.
L145 “We trained our model separately on Phylum, Class, Order, Family, Genus, and OTU”
Was this the same for the MetaSub dataset? Please specify
L236-238
Please correct repetitions and form
L242-244
It is not clear to me how the Authors selected these values for the Accuracy. I suppose that depends on a reasonable way to proceed, considering that predicting at the geographic level of cities is more difficult than for continents. However, it should be clarified.
L263-276
I suggest reorganising this section (or otherwise the Figure 2) in a way that the subsections of figure 2 are mentioned in ascending order.
L266
Figure 2D instead of 2C?
L279-282
Please correct the typo
L294-296 “Microbial species selected by Cassandra over multiple iterations were considered “bioindicator microbial species” for microbial prioritization, meaning that they were the top taxa unique or enriched to one location”
I would move and adapt this sentence in the intro section, at about L110, to specify what do you mean, in this specific context, with the term “bioindicator”.
L313-...
Being Results, I would avoid the words “interesting” and other subjective terms
L320 “and the feature ranks differed across selection for species at both continent and city levels”.
This is not so clear to me
L379 83%
This value is not mentioned in the Results section
L384-387
Please, rephrase the sentence to make it clear
L403 “TerristrialMetagenomeDB”
TerrestrialMetagenomeDB
L414 “While Cassandra can have myriad applications, proper interpretation necessitates the requirement of proper guidelines, orthogonal validation, and statistical ranking”
This sentence should be moved at the beginning of the discussion or at the enda t L421.
L470
Some of the abbreviations used are missing
In general, be consistent with the terminology (i.e. bio-indicator vs bioindicator, …)
Author Response
Thank you for your consideration of our manuscript, and we thank you for your patience as we prepared these revisions. We are grateful for the time you took to review and provide constructive comments. We have integrated your comments into our work and find it to be much improved. We are enclosing here an updated manuscript, and hope that it fully addresses the issues raised in your comments.
The reviewer's comments are in bold, while our replies to the reviewer are in blue. Updated sections based on comments are quoted inside “” with track changes. The new addition in track change mode is in green while deletion is denoted by strikethrough.

Reviewer 2 Report
Dear Authors,
your paper is well written and has a good structure.
Author Response
Thank you for your consideration of our manuscript, and we thank you for your time to review our manuscript.